# The GHG Emissions Generating Capacity by Productive Sectors in the EU: A SAM Analysis

**María T. Álvarez-Martínez** [1,*] and **Alfredo J. Mainar-Causapé** [2]

1   Fiscal Policy Analysis Unit, Joint Research Centre, 41092 Seville, Spain
2   Department of Applied Economics III, University of Seville, 41020 Seville, Spain; amainar@us.es
*   Correspondence: maria.alvarez@ec.europa.eu

**Abstract:** In this paper, we evaluate the generating capacity of Greenhouse Gases (GHG) emissions that all productive sectors have in the EU-27 of 2010. The analysis is performed using the social accounting matrices (SAMs) of each Member State (MS) and evaluating the interactions among industries, productive factors, and households with respect to the aggregated SAM for the EU-27. The main advantages and contributions of this study with respect to the existing literature are two. First, the availability of the whole income distribution detailed in the SAMs and second, their comparability across countries. The aim of this research is to better understand how productive sectors may damage the environment depending on their productive structure and final demand, particularly in a period of economic recession, which is very relevant in the context of COVID-19 and the near future. The results show that intersectoral connections are very diverse by MS and consequently, there are more differences in the generation capacity of GHG emission by country than by sector. Our results reinforce the idea of involving regional and national governments in the design and implementation of EU abatement strategies, taking into account the peculiarities of each region.

**Keywords:** social accounting matrices; structural decomposition analysis; European policy; GHG emissions; emission multipliers



## 1. Introduction

The European Union (EU) has played the role of global leader in the fight against climate change during the last decades. The ambitious plans of the EU to reduce green house gas (GHG) emissions have been registered in different strategies since the establishment of the European Climate Change Programme (ECCP), in 2000, in order to reach the goals of the 1997 Kyoto protocol. More recently, the 2015 Paris Climate Conference (COP21) strengthened the responsibility of countries to keep global warming below 2 °C compared to the average industrial times. The objective was to prevent severe weather impacts and catastrophic changes and reduce GHG emissions in 20% for 2020. The COP21 was the first universal legally binding global climate agreement and the EU has been since then at the forefront of these international efforts. In the period 2014–2020, the EU spent 20% of its budget on climate actions and the Commission has proposed to raise this share up to at least 25% in 2021–2027. Additionally, it has also economically contributed to support climate issues and renewable energy projects in developing countries.

In the European Green Deal, the European Commission (EC) proposed a reduction of GHG emissions of at least 55% for 2030 in relation to 1990 levels. This medium-term target is larger than the 20% in the 20-20-20 package and it is in line with the long run EC strategy of a climate-neutral society in 2050 (net-zero GHG). In order to achieve this goal, the EC has proposed the European Climate Law whereby each EU Member States (MS) would have to develop a binding national long-term strategy promoting fairness and solidarity among MS. It will include measuring and keeping track of progress by national governments adjusting actions accordingly. However, the path to zero GHG will entail

different economic efforts for MS. There is no doubt that environment and emissions are closely related with economic growth and production structure. This is particularly clear in periods of economic recessions, as it shows the EU with the highest cut down of GHG emissions in the period 2008–09 ($-7.2\%$, according to Eurostat). However, this aggregate reduction at the EU level hides different national production structures and economic growth rates and consequently, different macroeconomic impacts.

In the literature, there are studies that evaluate the trade-off between GHG mitigation and their corresponding costs. Most of the studies are at the national level, mainly due to the lack of comparable multi-country datasets. This issue has been recently fulfilled with the publication of the World Input-Output Database (WIOD), but the literature remains short yet. In general, these studies used the Structural Decomposition Analysis (SDA) or the index decomposition analysis (IDA). IDA is used to evaluate the drivers of energy use and emissions in a specific energy consumption sector and SDA is used by researchers who are more familiar with input–output (I–O) analysis and study changes in energy consumption and/or emissions in the whole economy [1]. These methodologies have been applied to several regional and national economies in the context of multi-regional studies and they are known as spatial structural decomposition analysis (see [2–4] on SDA). Spatial decomposition analysis can reveal differences in the performance of countries and regions in a wide context analysis. Proops et al. [5] did the earliest work. They studied the differences in $CO_2$ emissions between Germany and UK, and Chung [6] evaluated the differences for China, Japan and South Korea. De Nooij et al. [7] extended the scope to more countries and evaluated the differences on energy consumption among eight OECD countries while Hasegawa [8] did the comparison among regions in Japan.

There is scarce literature for the EU. Alcántara and Duarte [9] calculated the emission intensity differences among 14 EU countries using an SDA analysis performed with input-output data. They compared the results for each MS with the EU average, which was taken as reference economy. Duarte et al. [10] did an analysis for 10 EU countries and the US, focusing on the changes in GHG emissions associated to households' demand and evaluating the effects of technological factors (emission intensity and intermediate inputs substitution) and demand effects (consumption patterns, distribution of the demand, demand per capital and population). They conclude that technological change and efficiency improvements were not enough to compensate for the negative impact of economic growth. Brizga et al. [11] also found that economic growth was the main driver of GHG emissions in the Baltic countries (Estonia, Latvia, and Lithuania). More recently, Perrier et al. [12] elaborated an SDA using input-output tables for the 28 EU countries in the period 2009–2014. The EU emissions were broken down into five main drivers: population, consumption per capita, consumption, and production structure and carbon intensity of production. They conclude that although carbon intensity and economic recovery were the main drivers, the contribution of consumption and production structures and the temperature anomalies also play a major role.

However, all these analyses rely on input-output data and there are none international studies using social accounting matrices (SAMs). A SAM is a database that collects in a square form the economic and social data for all transactions between economic agents in a specific period of time, usually one year. It integrates social statistics in the traditional input-output table in such a way that the interdependence of productive and institutional sectors and their relationship with final demand are well captured through income flows. This closes the circular income flow in a square matrix (see [13,14], among others). There are studies that evaluate energy intensity and mitigation using a SAM but they have been performed at the national level (Pal and Pohit [15], Pal et al. [16] for India, Rodriguez et al. [17], Cansino et al. [18], Duarte et al. [19] for Spain) or regionally (Manresa and Sancho [20], Pié [21] for Catalonia, among others)

Our study looks to extend this literature on contributing to the comparative analysis of GHG emissions intensity internationally. It opens new discussions using as databases SAMs for each of the EU-27 MS. Consequently, this paper offers two significant contributions,

international comparability and complete circular flow of income effects. It evaluates the productive sectors that have the highest capacity of generating GHG emissions in each country and how they differ across EU MS. We compare the position of each sector in relation to the value of that sector in the EU-27, which is our reference economy. In particular, we try to identify the effects (own, direct, and indirect plus induce effects, which definition is in the following Section) that determine the differences from the average.

The rest of the paper is organized as follows. Section 2 describes the main linear SAM model with GHG emissions and a brief description of the database [22] and Section 3 presents the empirical application and the main results. Finally, Section 4 concludes and provides some policy recommendations.

## 2. Database and Methodology

In this Section, we mainly describe the procedure to evaluate the generating capacity of GHG emissions by sector in each EU-27 MS using emission multipliers. These multipliers are calculated using as the starting point a multisectoral linear input-output model applied to national SAMs. We combined data from emissions with the information included in SAMs.

### 2.1. The Dataset

The six gases included in the definition of GHG emissions are carbon dioxide, methane, nitrous oxide, perfluorocarbon, nitrogen trifluoride, and sulphur hexafluoride and their data come from *Air emissions accounts by NACE Rev. 2 activity (env_ac_ainah_r2)*, Eurostat. This data source provides information of the emissions in annual tons equivalent for all EU-27 MS (EU-27 in 2010 includes UK, but not Croatia). The sectoral information in Eurostat and national SAMs has been aggregated in order to match both data sources. The SAMs we use were done by Álvarez-Martínez and López-Cobo [22] for the EU-27 in 2010. The World Input–Output Database (WIOD) was the main data source ([23–25]), and they were completed with information from National Accounts in Eurostat. The SAMs include a disaggregation of labor by skill levels (high, medium, and low) and the disaggregation of the foreign sector into the EU and rest of the world (RoW). These matrices also include a great amount of detail on transfers and tax revenue allocations.

The SAMs of the EU-27 MS are balanced square matrices of dimension 85×85. There are four agents: households, the corporate sector, government, and the foreign sector, which are divided into the EU and the RoW. There are 59 productive sectors, nine accounts for wages and employers' social contributions by skill levels, social contributions paid by employees, self-employed and unemployed, and an account for capital. There are three accounts of taxes: direct taxes (households' income tax and corporate income tax), taxes net of subsidies on products and other taxes net of subsidies on production. Additionally, there is an account for property income and three types of transfers: other current transfers, adjustments due to the participation of households in pension funds reserves, and welfare benefits. Finally, there are two more accounts for gross fixed capital formation and stock variations, one for savings and three for trade and transport margins, international trade margins, and re-exports, correspondingly. The last three accounts are explicitly included in WIOD in order to match trade flows between countries and assure consistent flows within the EU and with the RoW (the name of the accounts and the structure of the SAMs are detailed in the Appendix A). All figures in the tables are at market prices and they are the most up-to-date set of compatible national matrices for Europe that have been benchmarked to publicly available official data (a more detailed description of the SAMs can be found in Álvarez-Martínez and López-Cobo [22]).

Finally, we have aggregated the 27 homogeneous SAMs into the SAM-EU27 that is used as reference in comparing all data and results in the analysis.

*2.2. The Model*

In this paper, the empirical application is performed with the SDA technique applied to sectoral production. The SDA allows a certain vector (or value) to be decomposed into a series of additive components. In the current case, we disentangle the vector of differences between emissions intensities generated by each MS and the EU-27 average into three main components, which aggregated resemble the total difference. These three components are the direct emissions intensity, the indirect effect, associated to the production process through the demand of intermediate inputs, and the induce effect, which is the result of the final demand and the infinite use of aggregated value added. Using this approach, we can identify groups of countries and sectors that stand out for their significant contribution to the generation capacity of GHG emissions.

Since this capacity of generating GHG emissions is evaluated thought the analysis of emission multipliers, we need to calculate first output multipliers [26] for each of the EU-27 MS and for their aggregate.

The starting point is the classical expression of the Leontief inverse extended to SAM models [26]. This is the traditional equilibrium equation:

$$\mathbf{x} = \mathbf{A}\mathbf{x} + \mathbf{y} \Leftrightarrow \mathbf{x} = (\mathbf{I} - \mathbf{A})^{-1}\mathbf{y} = \mathbf{M}\mathbf{y}, \tag{1}$$

where $\mathbf{x}$ is the vector of total gross output of endogenous accounts and $\mathbf{y}$ is the corresponding vector of total final demand. $\mathbf{A}$ is the technical coefficients matrix of the endogenous accounts in the SAMs (in this study, the exogenous accounts are: government, savings-investment, and rest of the world) and their components $a_{ij}$ account for the demand of sector $i$ needed to produce one unit of product $j$. $\mathbf{I}$ is the identity matrix and $\mathbf{M}$ is the matrix of SAM accounting multipliers. Although $\mathbf{A}$ and $\mathbf{M}$ are square matrices with the number of endogenous accounts, from now on $\mathbf{M}$ will be only referring to productive sectors in order to keep focus the analysis without changing notation and for simplicity:

$$\mathbf{M}(\mathrm{i}) = (\mathbf{I} - \mathbf{A}(\mathrm{i}))^{-1}. \tag{2}$$

In this expression, i = 1, . . . , 27 denotes the MS. Output multipliers are calculated as the aggregation of each column in matrix $\mathbf{M}$, which correspond to productive activities, and they show how output rises when there is an exogenous shock in one of the exogenous demands of any productive sector. We can pre-multiply $\mathbf{M}$ by the transposed vector of direct emission intensities $\mathbf{c}$, defined as GHG emissions (in tons equivalent of $CO_2$) per unit of output (thousands of euros) for each activity (this vector can be obtained using data from Eurostat, *env_ac_ainah_r2*). As result, we obtain a vector of total emissions, generated directly, indirectly, and induced, as per additional unit of demand in each productive sector. This is a vector of multipliers of total emissions generated by each productive sector in the economy:

$$\mathbf{e}'(\mathrm{i}) = \mathbf{c}\prime(\mathrm{i})\mathbf{M}(\mathrm{i}). \tag{3}$$

The study of vectors $\mathbf{e}(\mathrm{i})$ in the EU-27 MS and their direct comparison among them and in relation to the aggregated vector for the EU-27 can be used as a first analysis of the different capacities of generating emissions. However, it is the disaggregation of these differences in relation to the EU-27 that sheds light on the factors that explain them. These differences can be express as:

$$\Delta\mathbf{e}' = \mathbf{e}'(\mathrm{i}) - \mathbf{e}'(\mathrm{EU27\_i}) = \mathbf{c}'(\mathrm{i})\mathbf{M}(\mathrm{i}) - \mathbf{c}\prime(\mathrm{EU27\_i})\mathbf{M}(\mathrm{EU27\_i}). \tag{4}$$

This vector reflects the decomposition of $\Delta\mathbf{e}'$ in terms of $\Delta\mathbf{c}'$ and $\Delta\mathbf{M}$, with $\Delta$ meaning the differences between the vector for each EU-27 MS and the EU-27 aggregate. We obtain the contribution of each factor to the difference of sectoral capacity of generation among countries using the structural decomposition analysis (SDA). In this decomposition, the changes in a variable are determined by a series of multiplicative factors [27]. For example,

in expression $y = x_1 \cdot x_2$, SDA decompounds $\Delta y$ expressing how much of this variation is due to changes in $x_1$, how much is due to $x_2$, and which part is produced by the mixture of both. (For example, in $y = x_1 \cdot x_2$, it could be specified $\Delta y = x_1(t_1)\Delta x_2 + x_2(t_2)\Delta x_1$, where t1 and t2 refer to: two periods (initial vs. final); to different geographic areas or two economies (regions vs. average or national)). Obviously, SDA supports different variations depending on the assumptions made about the base period or regions used (see [6,8]).

In our decomposition, there are only two disaggregated factors (c as direct emissions intensity and M, the multiplier matrix), then we can perform, as in Dietzenbacher et al. [28], an exact decomposition as the following (indicating the average value between MS and the rest of EU27 MS as $\frac{1}{2}$):

$$\Delta \mathbf{e}' = \Delta \mathbf{c}' \mathbf{M} \left(\frac{1}{2}\right) + \mathbf{c}' \left(\frac{1}{2}\right) \Delta \mathbf{M}. \tag{5}$$

In this way, all differences among EU-27 MS are explained as the aggregation of differences in the direct intensity coefficients of emissions ($\Delta \mathbf{c}$) and the differences in production and distribution structures of each MS ($\Delta \mathbf{M}$). Nonetheless, there is even a further disaggregation because M is taken from SAMs, which accounts for the complete circular flow of income. This allows to distinguish between the effects due to the own productive structure of each country and the general effects derived from the distribution of income and value added. In order to reach this, we decompose matrices $\mathbf{M}$ in an additive way following the procedure proposed by Polo et al. [29]. They consider two groups of endogenous accounts in matrix $\mathbf{M}$. This procedure consists of decomposing the coefficient matrix of endogenous accounts, $\mathbf{A_n}$, in two matrices, $\mathbf{A_n} = \mathbf{B_1} + \mathbf{B_2}$, with $\mathbf{B_1}$ as a submatrix of productive sectors in rows and columns and zero otherwise, and $\mathbf{B_2}$ as a submatrix of all other endogenous accounts in $\mathbf{A_n}$. Defining $D = (\mathbf{I} - \mathbf{B_1})^{-1}\mathbf{B_2}$ and manipulating this expression properly, we arrive to $\mathbf{M} = \mathbf{M_3}\mathbf{M_2}\mathbf{M_1}$ which can be transformed into:

$$\mathbf{M} - \mathbf{I} = \mathbf{N_1} + \mathbf{N_2} + \mathbf{N_3}. \tag{6}$$

where $\mathbf{N_1}$ represents own net effects, which capture the effects of direct and indirect production exclusively needed by the productive sectors to satisfy a new unit of exogenous demand. $\mathbf{N_2}$ accounts for the indirect effects derived from the output needed to satisfy the induced demand of the endogenous accounts of productive sectors due to the own net effects. Finally, $\mathbf{N_3}$ accounts for all other effects, which are indirect effects (see also [30]).

We can reach a final expression to explain the differences in the generating capacity of emissions between EU-27 MS and the EU-27. First, we introduce expression (5) in (4) and take ($\mathbf{N_2} + \mathbf{N_3}$) as the indirect and induce net effects, then we take first differences in $\mathbf{N_1}$, which are direct net effects. As a result, the differences in multipliers ($\Delta \mathbf{e}'$) are explained by the following components:

$$\Delta \mathbf{c}' \mathbf{M} \left(\frac{1}{2}\right) \tag{7}$$

It is the share of differences due to direct intensities of emission in the productive process of each sector (differences based on direct effects).

$$\mathbf{c}' \left(\frac{1}{2}\right) \Delta \mathbf{N_1} \tag{8}$$

It is the share of differences due to the productive structure in a MS (differences based on own effects).

$$\mathbf{c}' \left(\frac{1}{2}\right) (\Delta \mathbf{N_2} + \Delta \mathbf{N_3}) \tag{9}$$

It is the share of differences due to the distribution structure of value added and the income generated the primary productive factors in a MS (differences based on circular effects).

## 3. Main Results and Discussion

In this Section, we present how an increase in the exogenous demand raises income, and consequently pollution, by type of sector in each MS. This is a key aspect in designing and structuring European economic policies aimed to reduce GHG emissions.

Table 1 presents the emission multipliers of broad sectors in all EU-27 MS. The values reported are those in vector e(i), which show how an exogenous and unilateral increase raises the pollution of each sector. More specifically, it reflects how many tons of $CO_2$ equivalent ($CO_2$ eq.) emissions are generated by each industry in each country when there is an increase of one million euro in exogenous accounts. In general, for the EU-27 aggregated and as it can be expected, the sector with the highest capacity of generation of GHG emissions is the energy sector, with 2847.1 tons of $CO_2$ eq., followed by agriculture (1313.3 tons of $CO_2$ eq.), and transport (980 tons of $CO_2$ eq.). Quite below are construction (423.5 tons of $CO_2$ eq.), manufactures (405.8 tons of $CO_2$ eq.), services (328.7 tons of $CO_2$ eq.), and mining (318.3 tons of $CO_2$ eq.). In general, this is the trend in most MS, although there are some exceptions that are worth to mention. The level of emissions generated by sectors increase in Estonia, Bulgaria, and Malta, where the difference in the generation capacity between energy and agriculture, with respect to the EU-27 aggregated, increases from 116.8% to 603.4%, 517% and 505%, correspondently. On the contrary, Ireland and France have bigger emission multipliers for agriculture than for energy, but with much lower differences, 19% and 6.8%.

**Table 1.** Emissions multipliers (tons of $CO_2$ eq. by million euro of output) of broad sectors in European Union Member States (2010).

| | Agriculture | Mining | Manufactures | Energy | Construction | Transport | Services | All Sectors |
|---|---|---|---|---|---|---|---|---|
| EU-27 | 1313.3 | 318.3 | 405.8 | 2847.1 | 423.5 | 980.2 | 328.7 | 399.7 |
| Austria | 866.2 | 149.2 | 205.7 | 1002.4 | 286.2 | 342.1 | 149.6 | 210.8 |
| Belgium | 801.1 | 44.4 | 182.9 | 1202.5 | 261.8 | 276.4 | 145.1 | 197.7 |
| Bulgaria | 1930.6 | 575.1 | 895.0 | 11,912.5 | 1209.6 | 1198.9 | 1229.4 | 1309.4 |
| Cyprus | 954.0 | 149.7 | 236.2 | 5995.8 | 774.8 | 345.2 | 507.9 | 464.5 |
| Czech Republic | 1518.4 | 1060.1 | 413.9 | 5886.7 | 618.6 | 662.5 | 564.4 | 548.7 |
| Germany | 986.8 | 203.4 | 272.6 | 3961.2 | 324.0 | 808.5 | 272.9 | 313.1 |
| Denmark | 1089.9 | 323.4 | 169.2 | 2656.1 | 235.3 | 1338.0 | 201.1 | 364.8 |
| Spain | 974.7 | 196.7 | 328.5 | 1576.7 | 342.3 | 782.4 | 247.3 | 324.5 |
| Estonia | 2200.1 | 1045.0 | 782.2 | 15,475.4 | 1033.5 | 1834.5 | 1199.8 | 1142.0 |
| Finland | 1138.5 | 118.1 | 402.3 | 3559.5 | 411.2 | 905.1 | 320.6 | 396.9 |
| France | 1148.5 | 69.5 | 211.0 | 1075.7 | 233.4 | 603.8 | 164.4 | 213.8 |
| United Kingdom | 1391.2 | 488.0 | 291.9 | 2981.7 | 353.0 | 1302.3 | 296.8 | 331.0 |
| Greece | 1194.2 | 244.6 | 435.7 | 6888.2 | 610.5 | 507.4 | 535.2 | 538.4 |
| Hungary | 1330.3 | 224.6 | 236.9 | 2566.9 | 419.3 | 686.7 | 364.9 | 342.8 |
| Ireland | 3608.8 | 251.3 | 203.7 | 3026.2 | 352.5 | 1229.0 | 157.3 | 224.4 |
| Italy | 720.2 | 63.0 | 294.0 | 1718.3 | 335.5 | 892.6 | 220.7 | 286.5 |
| Lithuania | 1947.8 | 51.9 | 501.6 | 2892.8 | 564.7 | 1457.8 | 581.4 | 678.9 |
| Luxembourg | 859.4 | 29.4 | 115.6 | 1254.1 | 144.2 | 1100.0 | 26.3 | 98.3 |
| Latvia | 2018.7 | 228.6 | 461.5 | 2184.1 | 726.9 | 1053.4 | 530.5 | 636.8 |
| Malta | 427.1 | 501.7 | 171.9 | 2583.9 | 361.6 | 2885.1 | 386.1 | 539.1 |
| Netherlands | 858.2 | 129.9 | 195.2 | 1942.1 | 178.2 | 824.4 | 163.1 | 225.3 |
| Poland | 2427.0 | 1386.2 | 816.9 | 7012.6 | 975.9 | 1324.4 | 1073.0 | 968.0 |
| Portugal | 1071.5 | 150.0 | 314.2 | 2447.2 | 589.4 | 750.7 | 307.4 | 384.6 |
| Romania | 2122.1 | 1440.7 | 752.2 | 5465.0 | 964.6 | 1311.4 | 942.8 | 956.9 |
| Slovakia | 800.5 | 227.1 | 445.8 | 1507.8 | 623.4 | 962.0 | 415.6 | 476.6 |
| Slovenia | 1283.0 | 742.5 | 283.5 | 3777.4 | 430.0 | 1365.4 | 350.9 | 402.0 |
| Sweden | 687.5 | 146.7 | 143.8 | 730.3 | 189.7 | 560.6 | 114.8 | 160.9 |

Source: Own elaboration.

If we focus attention on the transport sector, the size of these multipliers is also very different by MS. The countries above the EU-27 average are Bulgaria, Denmark, Estonia, UK, Ireland, Lithuania, Luxembourg, Latvia, Malta, Poland, Romania, and Slovenia. On the other side, the countries with the lowest capacity of generation are Belgium and Austria, two small countries. Looking at the more disaggregated sectors, the GHG emission generation capacity in the EU-27 is the highest for air transport followed by water transport,

inland and finally, other transport. Lithuania, Slovenia, Bulgaria, and Latvia are the MS with the highest GHG emission capacity of inland transport, while Luxembourg, Belgium, and Sweden are the MS with the lowest. In relation to air transport, the countries with the highest potential of generation are Luxembourg, UK, Ireland, Romania, and Latvia and the lowest are the Czech Republic, Slovakia, and Slovenia. In the case of the water transport, the countries with the highest potential of generating GHG emissions are Malta and far away, Estonia, Italia, and UK, in the opposite case are Austria, Luxembourg, and Slovenia. Thus, changes in transport policies should be evaluated at the country level, since the deployment of low-emissions alternative transports may have very different effects depending on the country. On this regard, as the EC is considering, local authorities will play a crucial role in the implementation of the European strategy to reduce transport emissions.

In general, the differences in the emission generation capacity of the same sector in two countries are due to the confluence of several factors, such as different production functions, different mix of inputs and different productive processes. The countries mainly depend on their technological development, but also on their environmental legislation and implementation, among other issues. Looking at the sectoral results in manufacturing, the results are very different by MS. The highest polluters are chemicals in Bulgaria, Latvia, and Romania, while this sector is the lowest polluter in Luxembourg. This is explained by the size of the sector in each country. Transport equipment has the highest capacity in Greece but not in other MS. In France, Germany, Italy, and Spain, there are not big differences among manufacture industries. These are the four biggest continental economies in the EU and the size of manufacturing is bigger than in other countries.

Finally, in the case of services, there are MS with high values such as Bulgaria, Czech Republic, Lithuania, etc., and others, where there are few differences among sectors. This study confirms the idea that there are more discrepancies by MS than by sectors. Some countries have high generation capacity of GHG emissions in all sectors, while others have lower generation capacity and similar values.

These results show the differences in the emission generation capacity by sector and country. As observed, there is not a common pattern on how emissions are generated in each sector. However, the figures of emission multipliers (Table 1) clearly show that some sectors are polluting more than others (agriculture, energy, and transport), and that some countries have an emission-generating capacity much higher than the rest, especially Eastern countries such as Bulgaria, Estonia, Poland, and Romania, where the industrial structure and technology are less developed. However, we should be cautious when considering industries and countries together since the size of each sector as a percentage of GDP may change our initial perception. An example is Ireland, with an important emission generation capacity, clearly above the average, in agriculture, energy, and transport, but with a high specialization in the service sector that makes its real contribution to the generation of emissions small compared to other MS.

On the other hand, as Table 2 shows, it is also relevant to understand how productive sectors pollute. In the case of agriculture, energy, and transport, which are the sectors with the highest value in total multipliers, the percentage of the direct effect is clearly predominant, around and above two thirds of the total. However, for manufacturing and construction, the use of inputs is the main cause of their polluting potential, being the circular effect of income the main cause in the services sector. This Table 2 reflects the sectoral emission multipliers and their decomposition in direct, own, and circular (indirect plus induced) effects for the EU-27 aggregate. The results should be read as follows. If agriculture is exogenously shocked by an increase of one million euros in-flow, the emissions of GHG increased by 1313.3 tons of $CO_2$ eq., of which 64.1% is due to the increase of output in the own sector, 22% due to direct effects on other sectors of the economy, and 13.9% due to interactions between other productive sectors. The results are different across sectors, as can be expected. Energy, transport, and agriculture have the highest direct effects and construction and manufactures, the highest own effects. Contrary, services is the sector with the highest circular effects. This is generally the case in all EU-27

MS. This table shows that indirect plus induced effects are the biggest contributors to emissions in the total economy because of services and to the high percentage in other sectors, while direct effects are the lowest contributors despite their high percentage in agriculture, energy, and transport. This is due to the relative smaller size of the sector in the whole economy.

**Table 2.** Emissions multipliers (tons of $CO_2$ eq. by million euro of output) of broad sectors in EU-27 and % decomposition in direct, own, and indirect plus induced effects.

| Sector | Me | cI | cN1 | cN2+cN3 |
| --- | --- | --- | --- | --- |
| Agriculture | **1313.3** | **64.1%** | 22.0% | 13.9% |
| Mining | **318.3** | **55.1%** | 24.3% | 20.6% |
| Manufactures | **405.8** | 16.5% | **48.6%** | 34.9% |
| Energy | **2847.1** | **72.7%** | 22.4% | 4.8% |
| Construction | **423.5** | 8.4% | **50.2%** | 41.4% |
| Transport | **980.2** | **64.9%** | 18.3% | 16.8% |
| Services | **328.7** | 7.9% | 29.6% | **62.5%** |
| Total (activities) | **399.7** | 17.5% | 38.3% | **44.3%** |
| Total (economy) | **353.5** | 12.8% | 28.1% | **59.1%** |

Source: Own elaboration. The bolded results are of significant value.

Looking into a wider sectoral disaggregation at the country level in Tables A1–A3 (see Appendix A), we can see that Luxembourg is the country with highest direct effects in fourteen sectors, Malta is the country with highest own effects in ten sectors, and Lithuania is the MS with the highest circular effects in eight sectors. This illustrates that the interconnection of sectors in Luxembourg and Lithuania contributes the most to generate emissions, while in Malta, these relations are less important. The detailed results are displayed in Tables A1–A3 in the Appendix A.

From the results in Tables 1 and 2 (and Tables A1–A3 in the Appendix A), it can be concluded that, when proposing policies aimed to reduce GHG emissions, it is necessary to track different sources and origins by sector and country. In this sense, Table 3 reinforces the idea that differences between countries are based fundamentally on the direct intensity of the production process. It is this component (**c**) that differs the most among MS and it is clearly determining the total difference (**e**). Table 3 contains the decomposition in GHG emission multipliers of EU MS vs. the EU-27 aggregate. The difference in emissions is divided into the variation due to direct intensities of emission (c), the differences in the productive structure (Ne1), and the indirect and induced effects due to the distribution structure of income and value added (Ne2,Ne3). As it is clearly stated in Table 3, the range of variation on differences between MS and the EU-27 is wide. Cyprus, UK, Finland, and Portugal are close to the EU-27 aggregate, while others such as Bulgaria, Estonia, and Poland are far away. The own effects are the leading force in emissions differentials and both open and circular effects, which refer to productive structures and distribution of income respectively, play a secondary role. Nonetheless, the circular effects dominate over the open effects.

**Table 3.** Decomposition (%) of differences in emissions multipliers of EU MS vs. rest of EU aggregate.

| EU Member States | Δe | Δc | ΔNe1 | ΔNe2 | ΔNe3 |
|---|---|---|---|---|---|
| Austria | −15.8% | −14.1% | 0.4% | 0.2% | −2.3% |
| Belgium | −17.5% | −3.5% | −5.7% | −0.9% | −7.4% |
| Bulgaria | 58.1% | 57.3% | 2.9% | 1.9% | −4.1% |
| Cyprus | −0.8% | 6.6% | −1.7% | −1.6% | −4.2% |
| Czech Republic | 12.7% | 16.3% | −2.6% | 1.3% | −2.3% |
| Germany | −5.0% | 3.5% | −3.2% | −1.4% | −3.9% |
| Denmark | −9.5% | 0.7% | −3.4% | −1.0% | −5.8% |
| Spain | −6.1% | −7.8% | 1.0% | 0.2% | 0.4% |
| Estonia | 53.8% | 60.6% | −0.1% | 1.7% | −8.4% |
| Finland | −1.5% | 3.5% | 0.2% | −0.4% | −4.7% |
| France | −13.6% | −10.4% | −1.4% | −0.3% | −1.6% |
| United Kingdom | −1.3% | 1.6% | −2.4% | −0.3% | −0.1% |
| Greece | 5.5% | 11.3% | −2.6% | −1.3% | −2.0% |
| Hungary | −4.3% | 1.7% | −2.4% | 0.6% | −4.3% |
| Ireland | −11.2% | 2.6% | −4.0% | −0.7% | −9.1% |
| Italy | −7.3% | −9.0% | 0.9% | 0.3% | 0.5% |
| Lithuania | 16.3% | 14.2% | −2.0% | 2.0% | 2.1% |
| Luxembourg | −13.5% | −3.8% | −1.7% | −0.3% | −7.6% |
| Latvia | 14.9% | 9.0% | 0.3% | 2.4% | 3.3% |
| Malta | 6.8% | 7.4% | 0.8% | 1.0% | −2.5% |
| Netherlands | −13.4% | −6.0% | −2.0% | −0.3% | −5.2% |
| Poland | 47.7% | 39.6% | 0.0% | 2.2% | 5.8% |
| Portugal | −2.3% | −4.4% | 0.8% | 0.7% | 0.6% |
| Romania | 41.1% | 24.3% | 3.7% | 3.3% | 9.9% |
| Slovakia | 5.3% | 1.2% | 0.0% | 2.1% | 1.9% |
| Slovenia | −2.3% | 5.0% | −2.7% | −0.3% | −4.3% |
| Sweden | −16.8% | −11.9% | −1.2% | −0.1% | −3.5% |

Source: Own elaboration.

## 4. Conclusions and Policy Recommendations

In this paper, we explore the energy intensities in the 27 EU MS and the EU-27 aggregate and their disaggregation into own, direct, and circular effects. The sectoral detail is wide, and we consider 27 productive sectors in each country. The analysis has been performed using the SDA technique applied to 27 EU SAMs elaborated by Álvarez-Martínez and López-Cobo [22]. This is the first time that the analysis is done with such a detailed homogenous information on the circular flow of income. This allows us to evaluate the effects that European economic policies oriented to the deployment of low-emission technologies may have in each EU MS and sector. The results show that these effects differ more by MS than by sector, a question that have not been evaluated up to now.

In general, for the EU-27 aggregate, the sector with the highest generation capacity is the energy sector. This is a common result also found in previous input-output analyses. However, we additionally compared these multipliers with the values found for other MS, and we found they are higher in Estonia, Bulgaria, and Malta while Ireland and France have them lower than for agriculture. The same happens for other sectors and countries such as transport and manufacture, since their effects are very different depending on the MS. This is reflecting the different levels of development in green technologies in each sector. For this reason, in order to reach the EU goals, the efforts done by each country must differ. There are countries where the generation capacity of GHG emissions is high in all sectors and there are other countries where the values are small and similar in all sectors.

Looking at the decomposition of sectoral emission multipliers into direct, own, and circular effects, the most significant are the circular effects due to the service sectors and their connections with other industries. However, again, it is worth to mention there are important differences across countries. The results show the intersectoral complexity of each economy and the different consequences that can be expected. In this sense, it is key for the elaboration of policies aimed at reducing GHG emissions to consider both the

different sources of emissions and the national industrial and technological structures. MS with a lower level of industrial development, but with and important share of the most contaminant sectors (agriculture, energy, or transport) in GDP, will require specific policies on the productive side of the economy, looking for a cleaner production. While in countries more services oriented, where GHG emissions are related with the circular flow of income, policies should be aimed to change consumption patterns. On this regard, a future line of research could be to extend our methodology to evaluate these relationships in order to clarify the way of formulating new GHG reduction policies. In all cases, it is very important to have in mind the sectoral interdependencies and the starting point of each economy to deal with EU objectives in terms of decarbonization. Consequently, the relevance of local, regional, and national governments in the implementation of green strategies to reduce emissions is a key factor.

**Author Contributions:** Conceptualization, M.T.Á.-M. and A.J.M.-C.; Formal analysis, M.T.Á.-M. and A.J.M.-C.; Investigation, M.T.Á.-M. and A.J.M.-C.; Methodology, A.J.M.-C.; Software, A.J.M.-C.; Supervision, M.T.Á.-M.; Writing—original draft, M.T.Á.-M. All authors have read and agreed to the published version of the manuscript.

**Funding:** The research received no external funding.

**Data Availability Statement:** Not applicable.

**Acknowledgments:** The authors thank two anonymous referees for their useful comments and suggestions. The views expressed are purely those of the authors and may not in any circumstances be regarded as stating an official position of the European Commission.

**Conflicts of Interest:** The authors declare no conflict of interest.

# Appendix A

**Table A1.** Direct effects (%).

| Sector | EU27 | AT | BE | BG | CY | CZ | DE | DK | ES | EE | FI | FR | UK | GR | HU | IE | IT | LT | LU | LV | MT | NL | PL | PT | RO | SK | SI | SE |
|---|---|---|---|---|---|---|---|---|---|---|---|---|---|---|---|---|---|---|---|---|---|---|---|---|---|---|---|---|
| Agriculture, hunting, forestry, and fishing | 64.1 | 71.1 | 83.8 | 43.8 | 49.5 | 56.4 | 74.9 | 73.2 | 66.7 | 43.2 | 64.3 | 71.7 | 76.2 | 49.7 | 61.4 | 80.1 | 57.9 | 64.1 | 90.6 | 63.7 | 45.8 | 71.2 | 49.3 | 65.6 | 45.8 | 47.7 | 68.0 | 80.5 |
| Mining and quarrying | 55.1 | 66.1 | 55.4 | 16.7 | 20.1 | 76.0 | 40.0 | 57.8 | 60.7 | 14.0 | 37.8 | 37.9 | 63.1 | 20.2 | 56.2 | 29.6 | 52.8 | 6.9 | 36.0 | 23.8 | 12.4 | 39.8 | 59.3 | 34.9 | 46.8 | 60.1 | 64.9 | 63.9 |
| Food, beverages, and tobacco | 8.3 | 14.6 | 17.2 | 4.8 | 8.7 | 7.8 | 10.0 | 12.8 | 6.5 | 6.1 | 4.0 | 11.6 | 11.7 | 2.9 | 8.5 | 7.6 | 9.9 | 8.7 | 8.1 | 9.5 | 3.2 | 14.9 | 6.3 | 11.6 | 4.4 | 17.9 | 8.7 | 12.5 |
| Textiles, leather, and footwear | 5.4 | 4.9 | 12.8 | 2.9 | 0.8 | 3.6 | 4.3 | 3.4 | 8.9 | 2.2 | 1.0 | 9.2 | 8.1 | 0.6 | 4.8 | 2.1 | 6.6 | 5.0 | 36.5 | 6.1 | 2.8 | 6.7 | 1.7 | 17.0 | 2.1 | 6.5 | 7.2 | 5.2 |
| Wood and cork | 9.3 | 11.8 | 5.6 | 7.0 | 8.5 | 3.6 | 10.7 | 7.9 | 23.0 | 3.1 | 2.4 | 8.2 | 20.8 | 1.9 | 9.4 | 8.9 | 11.7 | 17.0 | 44.5 | 14.9 | 7.9 | 7.1 | 6.8 | 23.2 | 3.3 | 11.0 | 4.3 | 7.7 |
| Pulp, paper, printing and publishing | 16.6 | 42.1 | 23.2 | 11.8 | 3.4 | 14.4 | 21.1 | 10.4 | 26.9 | 7.2 | 25.7 | 19.2 | 15.9 | 5.1 | 10.8 | 0.8 | 21.0 | 14.6 | 20.3 | 29.5 | 1.2 | 19.8 | 9.3 | 34.3 | 14.3 | 12.8 | 33.5 | 24.8 |
| Coke, refined petroleum, and nuclear fuel | 47.3 | 72.4 | 63.4 | 0.4 | 0.0 | 9.6 | 54.1 | 37.3 | 53.9 | 30.8 | 54.6 | 57.6 | 53.6 | 50.3 | 51.2 | 23.0 | 77.6 | 53.7 | 0.0 | 0.0 | 4.9 | 59.1 | 34.3 | 55.0 | 30.6 | 60.5 | 0.2 | 61.6 |
| Chemicals | 31.6 | 45.4 | 50.4 | 32.1 | 3.7 | 50.9 | 34.4 | 12.9 | 33.6 | 1.6 | 23.9 | 40.6 | 34.4 | 31.1 | 40.7 | 6.0 | 31.0 | 83.4 | 14.3 | 16.5 | 7.1 | 55.4 | 32.1 | 39.1 | 49.9 | 50.2 | 18.4 | 42.7 |
| Rubber and plastics | 10.6 | 3.2 | 11.1 | 9.5 | 6.2 | 3.2 | 12.9 | 7.3 | 14.8 | 2.7 | 2.1 | 14.7 | 25.4 | 23.3 | 6.7 | 13.7 | 9.8 | 1.9 | 21.9 | 7.0 | 4.5 | 10.1 | 3.7 | 7.3 | 7.3 | 42.2 | 12.0 | 14.1 |
| Other non-metallic mineral | 59.2 | 74.4 | 73.5 | 52.0 | 60.8 | 42.3 | 64.7 | 67.3 | 67.2 | 55.2 | 50.0 | 68.0 | 55.8 | 59.1 | 60.2 | 67.1 | 64.8 | 64.2 | 81.6 | 77.0 | 3.3 | 50.1 | 43.6 | 70.2 | 48.5 | 56.8 | 59.6 | 77.2 |
| Basic metals and fabricated metal | 30.4 | 56.3 | 47.7 | 6.0 | 7.4 | 33.6 | 34.0 | 14.9 | 34.4 | 3.5 | 43.8 | 39.3 | 43.0 | 11.6 | 38.2 | 54.7 | 30.3 | 3.0 | 57.7 | 40.2 | 4.3 | 51.8 | 17.8 | 10.0 | 29.1 | 63.0 | 20.0 | 51.4 |
| Machinery, nec | 4.1 | 4.5 | 21.8 | 0.8 | 2.3 | 2.8 | 5.6 | 9.4 | 7.9 | 2.1 | 0.5 | 3.9 | 6.5 | 4.5 | 4.9 | 10.7 | 5.1 | 0.7 | 11.0 | 2.4 | 5.4 | 6.7 | 1.3 | 17.0 | 2.0 | 11.8 | 3.1 | 4.2 |
| Electrical and optical equipment | 4.5 | 8.8 | 4.4 | 3.5 | 0.6 | 7.0 | 4.7 | 2.8 | 8.2 | 1.7 | 0.3 | 6.2 | 5.1 | 53.1 | 4.1 | 5.3 | 4.4 | 1.4 | 3.3 | 1.3 | 0.7 | 8.5 | 1.0 | 0.9 | 5.0 | 2.1 | 6.9 | 3.0 |
| Transport equipment | 4.0 | 1.8 | 12.3 | 0.7 | 0.3 | 2.8 | 5.5 | 4.9 | 10.9 | 1.5 | 1.1 | 4.6 | 7.0 | 1.4 | 5.8 | 2.1 | 4.2 | 0.7 | 2.6 | 3.1 | 2.8 | 4.9 | 1.5 | 1.5 | 1.2 | 6.5 | 3.9 | 5.5 |
| Manufacturing nec; recycling | 6.7 | 5.8 | 15.4 | 3.4 | 5.8 | 21.7 | 7.5 | 10.7 | 1.0 | 5.3 | 0.4 | 13.9 | 10.0 | 4.1 | 8.8 | 17.6 | 8.5 | 3.7 | 13.5 | 11.4 | 13.5 | 19.5 | 1.7 | 5.8 | 1.9 | 12.0 | 1.6 | 11.3 |
| Electricity, gas, and water supply | 72.7 | 41.8 | 88.0 | 87.2 | 95.0 | 75.6 | 81.6 | 94.0 | 66.1 | 83.8 | 90.7 | 66.6 | 67.0 | 86.3 | 83.5 | 67.8 | 84.3 | 77.7 | 81.7 | 71.7 | 81.6 | 71.5 | 85.1 | 52.5 | 60.1 | 53.2 | 78.1 | 86.0 |
| Construction | 8.4 | 21.8 | 16.3 | 4.8 | 6.3 | 11.0 | 11.4 | 23.1 | 1.6 | 4.8 | 12.0 | 12.9 | 13.6 | 5.7 | 16.1 | 5.0 | 8.6 | 5.3 | 22.1 | 7.8 | 3.8 | 20.3 | 1.7 | 12.3 | 5.3 | 20.1 | 2.7 | 24.3 |
| Sale, maintenance, and repair of motor vehicles | 10.9 | 14.2 | 32.5 | 3.4 | 7.2 | 4.6 | 10.1 | 21.5 | 16.7 | 4.3 | 2.1 | 32.7 | 10.6 | 0.9 | 32.2 | 20.0 | 4.1 | 8.9 | 46.9 | 9.8 | 9.0 | 20.3 | 7.7 | 10.7 | 18.8 | 7.5 | 0.0 | 46.0 |
| Wholesale trade and commission trade | 10.7 | 15.6 | 7.6 | 7.3 | 14.6 | 4.7 | 10.5 | 14.4 | 13.0 | 3.8 | 0.8 | 10.4 | 15.5 | 0.9 | 21.8 | 12.8 | 15.9 | 2.0 | 27.5 | 16.3 | 6.1 | 19.5 | 19.9 | 22.8 | 7.8 | 6.2 | 0.0 | 14.2 |
| Retail trade, except of motor vehicles | 14.7 | 24.2 | 93.3 | 3.7 | 25.0 | 3.7 | 17.3 | 11.0 | 14.7 | 6.9 | 12.3 | 35.3 | 19.1 | 2.5 | 20.1 | 25.5 | 4.9 | 4.8 | 43.5 | 12.5 | 4.1 | 25.5 | 10.7 | 15.4 | 16.6 | 6.8 | 0.0 | 13.6 |
| Hotels and restaurants | 7.8 | 14.9 | 20.2 | 4.3 | 9.6 | 4.4 | 15.9 | 9.2 | 2.9 | 4.4 | 8.6 | 16.7 | 9.0 | 6.8 | 11.1 | 8.8 | 7.0 | 4.8 | 14.6 | 7.0 | 0.9 | 22.0 | 6.1 | 10.7 | 3.6 | 23.6 | 12.1 | 4.8 |
| Other inland transport | 51.1 | 59.2 | 51.4 | 51.5 | 85.5 | 41.2 | 44.1 | 59.4 | 69.9 | 38.8 | 59.3 | 64.1 | 51.6 | 69.4 | 49.1 | 71.4 | 46.5 | 76.3 | 76.8 | 69.0 | 55.9 | 58.5 | 41.8 | 62.4 | 31.6 | 53.5 | 74.4 | 59.5 |
| Other water transport | 73.5 | 85.4 | 43.1 | 14.4 | 58.8 | 51.8 | 74.1 | 93.9 | 77.7 | 68.9 | 84.7 | 59.9 | 84.7 | 32.5 | 54.3 | 35.8 | 89.3 | 58.9 | 69.6 | 43.7 | 96.1 | 83.7 | 17.8 | 70.7 | 52.1 | 9.9 | 0.0 | 87.9 |
| Other air transport | 77.1 | 82.2 | 82.7 | 21.6 | 78.5 | 3.2 | 77.3 | 90.5 | 75.9 | 9.0 | 79.8 | 85.7 | 89.0 | 40.2 | 47.0 | 89.6 | 74.9 | 31.1 | 98.4 | 76.1 | 58.9 | 85.6 | 35.0 | 58.0 | 56.5 | 3.5 | 8.3 | 89.4 |
| Other supporting and aux. transport activities | 12.2 | 8.3 | 24.4 | 2.8 | 19.5 | 7.1 | 25.0 | 7.4 | 1.5 | 7.3 | 1.8 | 5.2 | 13.4 | 0.0 | 60.1 | 3.5 | 18.6 | 2.5 | 51.4 | 14.2 | 3.1 | 24.3 | 10.5 | 6.8 | 5.5 | 14.0 | 16.2 | 6.7 |
| Post and telecommunications | 11.3 | 16.2 | 20.2 | 1.6 | 4.1 | 5.1 | 24.5 | 11.4 | 8.9 | 3.3 | 4.2 | 14.5 | 13.2 | 3.2 | 34.9 | 12.3 | 4.1 | 4.5 | 39.8 | 8.6 | 7.1 | 15.2 | 6.9 | 4.9 | 8.8 | 33.1 | 3.8 | 16.7 |
| Financial intermediation | 3.2 | 4.2 | 8.2 | 1.2 | 1.6 | 0.8 | 3.8 | 2.2 | 1.6 | 1.4 | 8.4 | 3.8 | 0.2 | 2.6 | 14.5 | 3.1 | 2.1 | 2.3 | 14.6 | 1.8 | 1.2 | 6.3 | 15.9 | 4.0 | 5.2 | 3.6 | 1.4 | 5.2 |
| Real estate activities | 1.7 | 1.1 | 3.3 | 0.4 | 0.4 | 3.5 | 1.5 | 3.3 | 0.5 | 5.9 | 1.5 | 1.5 | 1.3 | 0.1 | 15.3 | 0.5 | 0.8 | 1.5 | 6.6 | 8.6 | 2.1 | 5.8 | 1.2 | 0.6 | 3.8 | 10.4 | 6.9 | 4.4 |
| Renting of m&eq and other business activities | 5.3 | 6.3 | 14.2 | 5.3 | 8.5 | 2.9 | 7.0 | 5.0 | 2.4 | 3.4 | 4.5 | 11.6 | 4.7 | 6.0 | 15.2 | 2.1 | 4.1 | 4.2 | 11.3 | 8.2 | 6.3 | 12.2 | 5.8 | 5.9 | 7.3 | 14.6 | 6.2 | 9.8 |
| Public admin and defense . . . | 10.0 | 9.9 | 26.6 | 1.8 | 5.4 | 2.9 | 12.8 | 8.3 | 11.2 | 13.0 | 11.6 | 16.8 | 13.2 | 9.2 | 19.9 | 14.0 | 7.1 | 5.2 | 14.9 | 13.1 | 4.3 | 13.9 | 11.5 | 13.5 | 7.5 | 14.6 | 1.2 | 13.8 |
| Education | 8.7 | 27.1 | 23.3 | 10.4 | 1.5 | 4.6 | 11.0 | 4.3 | 9.1 | 3.7 | 2.2 | 17.9 | 10.2 | 14.6 | 15.2 | 9.0 | 0.7 | 5.2 | 31.2 | 6.5 | 4.1 | 14.2 | 7.9 | 3.6 | 4.7 | 12.1 | 12.3 | 5.1 |
| Health and social work | 7.2 | 9.0 | 16.2 | 4.2 | 3.3 | 8.5 | 10.1 | 4.2 | 8.0 | 2.6 | 2.1 | 18.7 | 6.0 | 12.8 | 11.4 | 3.9 | 6.3 | 4.5 | 23.8 | 15.5 | 6.4 | 14.1 | 7.1 | 10.9 | 4.1 | 14.5 | 14.7 | 9.9 |
| Other community, social, and personal services | 7.0 | 16.8 | 17.4 | 1.7 | 5.3 | 5.3 | 12.0 | 4.9 | 6.6 | 3.2 | 3.3 | 14.7 | 7.1 | 4.0 | 6.9 | 5.3 | 4.6 | 3.4 | 30.6 | 3.5 | 1.1 | 9.7 | 4.4 | 4.8 | 4.5 | 20.1 | 8.3 | 7.8 |

The highlighted results are of significant value.

**Table A2.** Own effects (%).

| Sector | EU27 | AT | BE | BG | CY | CZ | DE | DK | ES | EE | FI | FR | UK | GR | HU | IE | IT | LT | LU | LV | MT | NL | PL | PT | RO | SK | SI | SE |
|---|---|---|---|---|---|---|---|---|---|---|---|---|---|---|---|---|---|---|---|---|---|---|---|---|---|---|---|---|
| Agriculture, hunting, forestry, and fishing | **22.0** | 22.4 | 11.8 | 36.3 | 32.3 | 25.7 | 15.4 | 20.6 | 17.0 | 35.8 | 22.8 | 20.7 | 13.0 | 23.0 | 27.3 | 16.5 | 25.4 | 23.5 | 8.4 | 23.6 | 33.6 | 23.3 | 27.5 | 21.3 | 30.5 | 28.5 | 20.1 | 13.4 |
| Mining and quarrying | **24.3** | 20.9 | 32.6 | 54.4 | 53.4 | 13.8 | 35.7 | 19.5 | 25.9 | 59.4 | 40.6 | 37.7 | 16.3 | 52.9 | 28.5 | 54.2 | 28.5 | 46.8 | 53.6 | 42.3 | 68.9 | 42.8 | 20.5 | 41.5 | 31.6 | 22.7 | 21.1 | 22.4 |
| Food, beverages, and tobacco | **62.5** | 66.5 | 67.0 | 66.9 | 58.5 | 56.6 | 63.4 | 69.9 | 66.9 | 66.7 | 68.7 | 68.3 | 52.6 | 55.0 | 68.9 | 80.4 | 62.9 | 57.5 | 82.6 | 56.7 | 67.4 | 69.6 | 56.5 | 61.6 | 51.2 | 49.0 | 60.8 | 68.9 |
| Textiles, leather, and footwear | **51.5** | 58.1 | 59.4 | 56.4 | 46.6 | 49.8 | 55.2 | 52.1 | 54.9 | 61.0 | 59.4 | 50.0 | 41.5 | 46.6 | 57.9 | 62.2 | 55.8 | 43.2 | 51.9 | 47.2 | 66.3 | 60.0 | 46.5 | 44.2 | 55.8 | 51.2 | 57.3 | |
| Wood and cork | **59.7** | 69.3 | 69.1 | 65.5 | 47.3 | 63.3 | 61.4 | 53.1 | 51.4 | 67.9 | 75.2 | 71.4 | 43.4 | 49.0 | 63.1 | 72.6 | 54.9 | 48.9 | 52.0 | 61.1 | 62.8 | 60.2 | 54.8 | 54.6 | 49.3 | 50.1 | 64.7 | 77.6 |
| Pulp, paper, printing, and publishing | **47.2** | 41.1 | 50.3 | 53.6 | 45.6 | 45.6 | 47.4 | 41.1 | 45.9 | 63.0 | 56.8 | 45.1 | 38.5 | 46.6 | 53.0 | 46.1 | 48.5 | 41.0 | 62.3 | 31.8 | 61.5 | 48.3 | 45.0 | 41.1 | 39.8 | 53.2 | 42.9 | 57.1 |
| Coke, refined petroleum, and nuclear fuel | **37.4** | 19.3 | 29.6 | 73.6 | 45.0 | 68.2 | 33.3 | 45.7 | 35.6 | 51.7 | 34.7 | 29.6 | 33.1 | 26.6 | 34.0 | 55.2 | 15.7 | 25.3 | 79.9 | 53.6 | 66.1 | 33.4 | 41.7 | 29.1 | 45.4 | 30.0 | 54.6 | 29.9 |
| Chemicals | **42.0** | 37.2 | 37.7 | 52.1 | 46.8 | 33.6 | 39.5 | 47.7 | 44.0 | 70.0 | 51.9 | 38.7 | 33.8 | 33.9 | 43.4 | 46.3 | 47.9 | 9.1 | 67.6 | 42.4 | 59.3 | 35.0 | 40.1 | 36.5 | 34.4 | 34.9 | 48.8 | 36.3 |
| Rubber and plastics | **54.2** | 66.9 | 63.6 | 64.7 | 62.2 | 60.7 | 53.4 | 48.7 | 56.4 | 61.1 | 63.1 | 55.2 | 42.1 | 49.0 | 62.5 | 57.1 | 60.0 | 49.0 | 66.9 | 49.9 | 62.5 | 64.7 | 53.1 | 56.4 | 46.0 | 39.2 | 52.4 | 55.1 |
| Other non-metallic mineral | **27.8** | 19.3 | 21.7 | 36.8 | 33.1 | 37.1 | 24.1 | 19.6 | 24.5 | 29.8 | 33.4 | 22.1 | 27.1 | 28.7 | 28.8 | 25.8 | 26.2 | 18.6 | 17.4 | 14.9 | 65.1 | 34.2 | 33.3 | 22.5 | 33.3 | 29.1 | 27.2 | 17.6 |
| Basic metals and fabricated metal | **44.7** | 34.9 | 41.7 | 69.2 | 61.1 | 42.9 | 44.8 | 42.9 | 44.7 | 58.1 | 41.5 | 39.9 | 34.6 | 62.8 | 43.5 | 34.1 | 45.1 | 44.4 | 38.4 | 38.2 | 63.5 | 33.2 | 46.1 | 52.8 | 47.9 | 27.5 | 53.4 | 35.9 |
| Machinery, nec | **52.1** | 62.3 | 52.9 | 67.6 | 49.3 | 56.2 | 50.5 | 43.4 | 56.4 | 56.4 | 59.0 | 53.4 | 50.1 | 43.5 | 55.9 | 57.6 | 57.3 | 47.0 | 71.7 | 49.0 | 65.7 | 55.8 | 47.7 | 42.0 | 52.7 | 62.3 | 56.7 | 61.5 |
| Electrical and optical equipment | **49.7** | 56.7 | 60.1 | 60.6 | 47.0 | 44.3 | 48.6 | 47.1 | 57.2 | 52.1 | 52.0 | 54.4 | 42.7 | 21.8 | 55.8 | 52.4 | 55.5 | 47.0 | 78.7 | 48.3 | 67.4 | 56.4 | 50.3 | 47.1 | 44.8 | 66.1 | 51.3 | 55.9 |
| Transport equipment | **54.0** | 64.5 | 58.9 | 64.0 | 45.1 | 50.8 | 55.6 | 52.6 | 56.6 | 58.5 | 58.0 | 55.2 | 48.6 | 40.7 | 56.2 | 62.3 | 59.8 | 48.2 | 82.3 | 51.5 | 30.0 | 60.1 | 50.6 | 52.5 | 50.6 | 59.7 | 56.6 | 63.2 |
| Manufacturing nec; recycling | **51.6** | 65.1 | 59.0 | 63.3 | 48.3 | 41.6 | 52.4 | 45.9 | 61.9 | 59.9 | 62.6 | 50.9 | 43.0 | 47.4 | 55.3 | 58.5 | 53.9 | 44.7 | 70.3 | 51.6 | 51.0 | 47.9 | 49.9 | 55.0 | 47.5 | 51.1 | 56.9 | 56.8 |
| Electricity, gas, and water supply | **22.4** | 53.8 | 9.4 | 9.6 | 2.5 | 20.1 | 15.6 | 3.9 | 27.8 | 14.0 | 6.6 | 26.7 | 28.2 | 10.0 | 12.7 | 29.6 | 11.6 | 13.9 | 17.5 | 18.2 | 15.2 | 26.3 | 8.1 | 41.8 | 32.6 | 37.1 | 18.4 | 9.1 |
| Construction | **50.2** | 56.6 | 63.8 | 61.6 | 63.1 | 43.0 | 49.9 | 39.5 | 61.3 | 49.8 | 54.4 | 48.5 | 36.6 | 55.2 | 51.9 | 50.9 | 56.0 | 37.7 | 66.3 | 53.4 | 63.6 | 44.2 | 45.9 | 59.8 | 44.0 | 50.6 | 58.6 | 49.5 |
| Sale, maintenance, and repair of motor vehicles | **36.1** | 38.7 | 39.2 | 54.1 | 36.9 | 38.2 | 37.6 | 31.6 | 42.7 | 47.2 | 55.5 | 21.3 | 29.1 | 45.1 | 29.6 | 36.3 | 45.3 | 29.8 | 34.3 | 30.0 | 50.9 | 41.8 | 28.4 | 29.0 | 50.0 | 41.6 | 47.4 | 27.0 |
| Wholesale trade and commission trade | **38.9** | 43.0 | 55.8 | 52.8 | 30.1 | 44.0 | 43.9 | 40.8 | 47.2 | 45.5 | 57.9 | 32.9 | 30.3 | 38.5 | 42.4 | 54.4 | 39.8 | 33.4 | 42.1 | 33.9 | 65.3 | 40.6 | 30.1 | 32.0 | 25.1 | 43.6 | 49.9 | 42.9 |
| Retail trade, except of motor vehicles | **35.2** | 34.5 | 3.6 | 55.5 | 20.3 | 40.8 | 38.1 | 38.5 | 36.5 | 49.8 | 47.7 | 20.3 | 25.3 | 31.8 | 38.9 | 40.3 | 49.8 | 29.0 | 41.6 | 31.9 | 65.4 | 36.5 | 35.4 | 35.4 | 25.6 | 38.6 | 49.8 | 43.2 |
| Hotels and restaurants | **46.5** | 45.9 | 52.6 | 49.6 | 53.4 | 41.6 | 43.1 | 57.6 | 45.4 | 61.6 | 51.1 | 46.2 | 40.2 | 42.4 | 52.9 | 60.0 | 51.9 | 30.1 | 71.1 | 38.9 | 73.3 | 53.3 | 43.0 | 45.8 | 45.8 | 35.7 | 49.3 | 59.1 |
| Other inland transport | **24.0** | 24.8 | 32.7 | 28.1 | 14.7 | 29.0 | 30.6 | 26.7 | 16.5 | 33.1 | 23.4 | 16.2 | 17.9 | 13.4 | 31.8 | 16.1 | 26.0 | 8.7 | 13.8 | 16.0 | 30.3 | 22.8 | 27.7 | 22.8 | 29.1 | 29.7 | 16.9 | 25.7 |
| Other water transport | **14.2** | 8.3 | 38.9 | 39.0 | 28.7 | 20.1 | 16.0 | 3.4 | 13.4 | 17.3 | 8.8 | 15.9 | 6.5 | 18.6 | 30.0 | 40.3 | 6.2 | 13.2 | 15.5 | 21.8 | 3.5 | 12.8 | 37.1 | 15.5 | 24.0 | 43.5 | 32.1 | 10.2 |
| Other air transport | **12.6** | 12.6 | 14.8 | 36.1 | 9.6 | 49.7 | 13.8 | 6.4 | 14.3 | 54.2 | 12.1 | 8.4 | 4.8 | 22.3 | 35.9 | 6.2 | 15.5 | 32.0 | 1.1 | 10.7 | 28.7 | 10.7 | 29.3 | 22.6 | 21.0 | 62.2 | 34.5 | 7.4 |
| Other supporting and aux. transport activities | **45.2** | 54.3 | 55.0 | 56.4 | 52.5 | 40.8 | 57.9 | 63.8 | 61.9 | 69.4 | 34.8 | 24.3 | 52.5 | 19.2 | 72.8 | 41.1 | 34.0 | 26.1 | 46.8 | 50.4 | 49.5 | 44.4 | 44.2 | 16.0 | 54.5 | 53.1 | | 75.5 |
| Post and telecommunications | **33.1** | 38.0 | 41.7 | 50.6 | 39.4 | 26.2 | 32.3 | 39.9 | 46.0 | 38.3 | 50.7 | 27.6 | 27.4 | 24.9 | 24.3 | 39.7 | 38.9 | 19.7 | 34.4 | 29.3 | 55.9 | 35.4 | 26.8 | 29.3 | 23.6 | 28.3 | 33.7 | 41.8 |
| Financial intermediation | **24.3** | 33.8 | 30.9 | 32.6 | 19.1 | 20.5 | 26.0 | 21.7 | 25.4 | 31.2 | 43.9 | 20.8 | 24.9 | 20.8 | 27.2 | 27.8 | 20.7 | 23.9 | 48.0 | 15.3 | 49.6 | 29.6 | 23.9 | 18.4 | 36.9 | 25.9 | 21.7 | 30.8 |
| Real estate activities | **27.3** | 47.7 | 45.4 | 39.7 | 30.0 | 48.0 | 21.5 | 27.0 | 27.0 | 42.2 | 67.4 | 10.4 | 16.2 | 15.2 | 36.6 | 51.8 | 12.9 | 27.7 | 39.8 | 42.6 | 60.0 | 42.9 | 63.6 | 20.1 | 41.9 | 47.1 | 35.5 | 57.5 |
| Renting of m&eq and other business activities | **28.9** | 41.6 | 38.8 | 46.9 | 23.5 | 33.3 | 25.3 | 35.6 | 38.7 | 33.8 | 44.2 | 24.2 | 19.2 | 32.2 | 31.5 | 40.3 | 38.4 | 24.4 | 47.9 | 26.5 | 51.8 | 37.2 | 28.8 | 32.8 | 43.9 | 35.0 | 32.7 | 41.4 |
| Public admin and defence . . . | **30.3** | 36.4 | 24.4 | 38.4 | 40.7 | 26.7 | 32.2 | 31.0 | 34.6 | 41.8 | 42.4 | 20.4 | 25.7 | 43.2 | 30.4 | 29.4 | 32.0 | 26.0 | 54.9 | 27.0 | 46.7 | 43.0 | 19.2 | 31.9 | 23.3 | 32.2 | 32.4 | 40.6 |
| Education | **23.2** | 23.6 | 18.8 | 40.2 | 47.3 | 39.4 | 30.9 | 36.5 | 19.6 | 51.9 | 43.7 | 17.9 | 18.1 | 9.6 | 28.9 | 27.7 | 17.2 | 33.5 | 25.3 | 24.4 | 20.9 | 31.9 | 25.8 | 19.1 | 23.6 | 34.3 | 27.4 | 37.3 |
| Health and social work | **31.5** | 41.0 | 39.0 | 48.3 | 33.8 | 38.8 | 31.4 | 35.2 | 30.2 | 47.9 | 40.0 | 18.4 | 34.9 | 31.7 | 42.7 | 28.2 | 33.4 | 44.5 | 40.9 | 24.5 | 29.9 | 35.2 | 33.4 | 33.3 | 46.3 | 38.7 | 29.4 | 28.4 |
| Other community, social, and personal services | **34.9** | 41.2 | 42.0 | 58.2 | 35.3 | 50.0 | 31.8 | 42.6 | 38.9 | 56.8 | 55.1 | 29.2 | 22.7 | 28.6 | 45.5 | 37.6 | 33.4 | 28.6 | 34.7 | 35.9 | 78.1 | 57.5 | 41.2 | 45.0 | 42.3 | 38.5 | 45.9 | 49.4 |

The highlighted results are of significant value.

**Table A3.** Circular effects (%).

| Sector | EU27 | AT | BE | BG | CY | CZ | DE | DK | ES | EE | FI | FR | UK | GR | HU | IE | IT | LT | LU | LV | MT | NL | PL | PT | RO | SK | SI | SE |
|---|---|---|---|---|---|---|---|---|---|---|---|---|---|---|---|---|---|---|---|---|---|---|---|---|---|---|---|---|
| Agriculture, hunting, forestry, and fishing | **13.9** | 6.5 | 4.4 | 19.8 | 18.2 | 17.9 | 9.8 | 6.2 | 16.3 | 21.0 | 12.9 | 7.6 | 10.8 | 27.2 | 11.3 | 3.5 | 16.7 | 12.4 | 1.0 | 12.6 | 20.6 | 5.5 | 23.2 | 13.1 | 23.6 | 23.8 | 11.9 | 6.1 |
| Mining and quarrying | **20.6** | 13.0 | 12.0 | 29.0 | 26.5 | 10.2 | 24.3 | 22.8 | 13.4 | 26.6 | 21.6 | 24.5 | 20.6 | 26.9 | 15.2 | 16.2 | 18.7 | 46.4 | 10.4 | 33.9 | 18.8 | 17.4 | 20.2 | 23.6 | 21.5 | 17.2 | 14.0 | 13.7 |
| Food, beverages, and tobacco | **29.2** | 18.9 | 15.8 | 28.3 | 32.8 | 35.6 | 26.6 | 17.3 | 26.6 | 27.2 | 27.3 | 20.1 | 35.7 | 42.1 | 22.6 | 11.9 | 27.2 | 33.9 | 9.3 | 33.8 | 29.4 | 15.5 | 37.2 | 26.8 | 44.4 | 33.1 | 30.5 | 18.6 |
| Textiles, leather, and footwear | **43.1** | 37.1 | 27.8 | 40.7 | 52.6 | 46.7 | 40.4 | 44.4 | 36.2 | 36.8 | 39.6 | 40.8 | 50.4 | 52.8 | 37.4 | 35.7 | 37.6 | 51.8 | 11.5 | 46.8 | 31.0 | 33.4 | 51.8 | 38.7 | 53.6 | 37.8 | 41.6 | 37.5 |
| Wood and cork | **31.0** | 18.9 | 25.3 | 27.5 | 44.1 | 33.1 | 27.9 | 39.0 | 25.6 | 29.0 | 22.4 | 20.5 | 35.8 | 49.2 | 27.4 | 18.4 | 33.4 | 34.1 | 3.5 | 24.0 | 29.3 | 32.7 | 38.3 | 22.2 | 47.4 | 38.9 | 31.0 | 14.8 |
| Pulp, paper, printing, and publishing | **36.2** | 16.7 | 26.5 | 34.6 | 50.9 | 40.0 | 31.5 | 48.6 | 27.2 | 29.8 | 17.5 | 35.8 | 45.5 | 48.3 | 36.2 | 53.1 | 30.4 | 44.4 | 17.4 | 38.7 | 37.3 | 31.8 | 45.6 | 24.6 | 46.0 | 34.0 | 23.6 | 18.0 |
| Coke, refined petroleum, and nuclear fuel | **15.2** | 8.4 | 7.0 | 26.0 | 55.0 | 22.2 | 12.5 | 17.0 | 10.4 | 17.4 | 10.6 | 12.7 | 13.3 | 23.0 | 14.8 | 21.8 | 6.7 | 20.9 | 20.1 | 46.4 | 29.0 | 7.5 | 24.0 | 15.9 | 24.0 | 9.4 | 45.2 | 8.4 |
| Chemicals | **26.4** | 17.3 | 11.8 | 15.8 | 49.5 | 15.5 | 26.1 | 39.4 | 22.4 | 28.3 | 24.3 | 20.6 | 31.9 | 35.0 | 15.8 | 47.7 | 21.1 | 7.5 | 18.1 | 41.1 | 33.6 | 9.6 | 27.8 | 24.4 | 15.7 | 15.0 | 32.8 | 21.1 |
| Rubber and plastics | **35.1** | 29.8 | 25.3 | 25.8 | 31.6 | 36.2 | 33.7 | 44.0 | 28.7 | 36.3 | 34.8 | 30.1 | 32.5 | 27.7 | 30.9 | 29.1 | 30.2 | 49.1 | 11.1 | 43.1 | 33.0 | 25.3 | 43.2 | 36.3 | 46.6 | 18.7 | 35.5 | 30.8 |
| Other non-metallic mineral | **13.0** | 6.3 | 4.9 | 11.1 | 6.1 | 20.6 | 11.1 | 13.0 | 8.3 | 15.0 | 16.6 | 9.9 | 17.1 | 12.3 | 11.1 | 7.0 | 9.0 | 17.2 | 1.0 | 8.1 | 31.6 | 15.7 | 23.1 | 7.3 | 18.2 | 14.0 | 13.2 | 5.2 |
| Basic metals and fabricated metal | **24.9** | 8.8 | 10.7 | 24.8 | 31.5 | 23.4 | 21.2 | 42.2 | 20.9 | 38.4 | 14.8 | 20.8 | 22.4 | 25.6 | 18.3 | 11.2 | 24.6 | 52.5 | 4.0 | 21.7 | 32.2 | 14.9 | 36.0 | 37.1 | 23.0 | 9.5 | 26.5 | 12.7 |
| Machinery, nec | **43.8** | 33.2 | 25.3 | 31.6 | 48.4 | 41.0 | 44.0 | 47.2 | 35.7 | 41.5 | 40.5 | 42.7 | 43.3 | 52.0 | 39.2 | 31.6 | 37.6 | 52.3 | 17.4 | 48.6 | 28.9 | 37.5 | 50.9 | 41.0 | 45.2 | 25.9 | 40.2 | 34.3 |
| Electrical and optical equipment | **45.8** | 34.5 | 35.5 | 35.9 | 52.4 | 48.8 | 46.6 | 50.1 | 34.5 | 46.2 | 47.7 | 39.4 | 52.2 | 25.1 | 40.1 | 42.4 | 40.1 | 51.6 | 17.9 | 50.4 | 31.9 | 35.1 | 48.7 | 52.0 | 50.1 | 31.9 | 41.8 | 41.1 |
| Transport equipment | **42.0** | 33.7 | 28.9 | 35.3 | 54.6 | 46.5 | 38.9 | 42.5 | 32.5 | 40.0 | 41.0 | 40.1 | 44.5 | 57.9 | 38.0 | 35.7 | 36.0 | 51.1 | 15.1 | 45.4 | 67.2 | 35.0 | 47.9 | 46.1 | 48.2 | 33.8 | 39.4 | 31.3 |
| Manufacturing nec; recycling | **41.7** | 29.2 | 25.6 | 33.3 | 45.9 | 36.7 | 40.1 | 43.4 | 37.1 | 34.8 | 37.0 | 35.2 | 47.0 | 48.4 | 36.0 | 23.9 | 37.7 | 51.6 | 16.3 | 37.0 | 35.5 | 32.6 | 48.4 | 39.2 | 50.6 | 36.8 | 41.5 | 31.9 |
| Electricity, gas, and water supply | **4.8** | 4.4 | 2.7 | 3.2 | 2.5 | 4.3 | 2.8 | 2.1 | 6.2 | 2.2 | 2.7 | 6.7 | 4.8 | 3.6 | 3.8 | 2.6 | 4.1 | 8.4 | 0.8 | 10.1 | 3.2 | 2.2 | 6.7 | 5.8 | 7.3 | 9.7 | 3.5 | 4.9 |
| Construction | **41.4** | 21.6 | 19.9 | 33.6 | 30.6 | 46.0 | 38.7 | 37.5 | 37.2 | 45.4 | 33.5 | 38.6 | 49.9 | 39.2 | 32.0 | 44.1 | 35.4 | 57.0 | 11.6 | 38.7 | 32.6 | 35.5 | 52.4 | 27.8 | 50.8 | 29.3 | 38.6 | 26.1 |
| Sale, maintenance, and repair of motor vehicles | **53.0** | 47.2 | 28.3 | 42.5 | 55.9 | 57.2 | 52.3 | 46.9 | 40.6 | 48.5 | 42.4 | 46.0 | 60.3 | 54.0 | 38.2 | 43.6 | 50.5 | 61.3 | 18.8 | 60.2 | 40.1 | 37.9 | 63.9 | 60.4 | 31.2 | 50.9 | 52.6 | 27.0 |
| Wholesale trade and commission trade, | **50.5** | 41.3 | 36.7 | 39.9 | 55.3 | 51.3 | 45.6 | 44.8 | 39.9 | 50.7 | 41.3 | 56.7 | 54.2 | 60.6 | 35.8 | 32.7 | 44.3 | 64.6 | 30.3 | 49.8 | 28.6 | 39.8 | 50.0 | 45.2 | 67.1 | 50.2 | 50.1 | 42.9 |
| Retail trade, except of motor vehicles | **50.1** | 41.3 | 3.1 | 40.8 | 54.6 | 55.5 | 44.6 | 50.4 | 48.7 | 43.3 | 39.9 | 44.5 | 55.6 | 65.8 | 41.0 | 34.2 | 45.3 | 66.3 | 14.9 | 55.5 | 30.6 | 38.0 | 53.9 | 49.2 | 57.8 | 54.7 | 50.2 | 43.2 |
| Hotels and restaurants | **45.8** | 39.2 | 27.2 | 46.1 | 37.0 | 54.0 | 41.0 | 33.2 | 51.8 | 33.9 | 40.3 | 37.1 | 50.8 | 50.7 | 35.9 | 31.2 | 42.2 | 63.3 | 14.3 | 54.1 | 25.8 | 24.8 | 50.9 | 40.5 | 50.6 | 40.7 | 38.6 | 36.1 |
| Other inland transport | **24.9** | 16.0 | 15.9 | 20.4 | −0.3 | 29.8 | 25.3 | 14.0 | 13.6 | 28.1 | 17.3 | 19.7 | 30.5 | 17.2 | 19.1 | 12.5 | 27.5 | 15.0 | 9.4 | 15.0 | 13.9 | 18.7 | 30.5 | 14.8 | 39.3 | 16.8 | 8.7 | 14.8 |
| Other water transport | **12.3** | 6.3 | 18.0 | 46.6 | 12.6 | 28.1 | 10.0 | 2.6 | 8.9 | 13.9 | 6.5 | 24.3 | 8.8 | 48.9 | 15.7 | 23.8 | 4.5 | 27.9 | 14.9 | 34.5 | 0.4 | 3.5 | 45.2 | 13.8 | 23.9 | 46.6 | 67.9 | 1.9 |
| Other air transport | **10.3** | 5.3 | 2.5 | 42.2 | 11.9 | 47.0 | 9.0 | 3.2 | 9.8 | 36.8 | 8.0 | 5.9 | 6.3 | 37.5 | 17.1 | 4.2 | 9.7 | 36.9 | 0.6 | 13.2 | 12.4 | 3.7 | 35.7 | 19.4 | 22.5 | 34.3 | 57.2 | 3.2 |
| Other supporting and aux. transport activities | **42.6** | 37.4 | 20.7 | 40.8 | 28.0 | 56.3 | 34.2 | 34.6 | 38.3 | 28.8 | 60.0 | 62.3 | 47.5 | 20.6 | 23.7 | 40.3 | 63.5 | 22.5 | 39.1 | 46.6 | 26.3 | 45.1 | 49.0 | 78.5 | 31.5 | 30.7 | | 17.8 |
| Post and telecommunications | **55.6** | 45.9 | 38.1 | 47.7 | 56.5 | 68.8 | 43.3 | 48.7 | 45.1 | 58.5 | 45.1 | 57.9 | 59.4 | 71.9 | 40.7 | 47.9 | 57.1 | 75.7 | 25.8 | 62.1 | 37.0 | 49.4 | 66.3 | 65.8 | 67.6 | 38.6 | 62.5 | 41.5 |
| Financial intermediation | **72.5** | 62.0 | 60.9 | 66.1 | 79.4 | 78.7 | 70.1 | 76.0 | 72.9 | 67.4 | 47.7 | 75.4 | 74.9 | 76.6 | 58.3 | 69.1 | 77.2 | 73.8 | 37.4 | 82.9 | 49.2 | 64.1 | 60.2 | 77.6 | 57.9 | 70.5 | 76.9 | 64.0 |
| Real estate activities | **71.0** | 51.1 | 51.2 | 59.9 | 69.6 | 48.6 | 77.0 | 69.7 | 72.5 | 52.0 | 31.1 | 88.1 | 82.5 | 84.7 | 48.1 | 47.6 | 86.2 | 70.8 | 53.7 | 48.8 | 37.9 | 51.3 | 35.2 | 79.2 | 54.3 | 42.5 | 57.7 | 38.1 |
| Renting of m&eq and other business activities | **65.9** | 52.1 | 47.0 | 47.8 | 68.0 | 63.7 | 67.7 | 59.4 | 58.9 | 62.9 | 51.3 | 64.2 | 76.1 | 61.8 | 53.3 | 57.6 | 57.5 | 71.4 | 40.8 | 65.4 | 42.0 | 50.7 | 65.4 | 61.3 | 48.9 | 48.1 | 61.0 | 48.8 |
| Public admin and defense . . . | **59.6** | 53.7 | 49.0 | 59.9 | 53.9 | 70.4 | 55.0 | 60.7 | 54.2 | 45.2 | 46.0 | 62.9 | 61.1 | 47.5 | 49.7 | 56.6 | 61.0 | 68.8 | 30.2 | 59.8 | 49.0 | 43.1 | 69.3 | 54.6 | 69.1 | 53.2 | 66.4 | 45.7 |
| Education | **68.0** | 49.3 | 57.9 | 49.3 | 51.2 | 56.0 | 58.1 | 59.3 | 71.3 | 44.4 | 54.2 | 64.2 | 71.8 | 75.8 | 55.8 | 63.2 | 82.0 | 61.3 | 43.4 | 69.1 | 75.0 | 54.0 | 66.2 | 77.3 | 71.7 | 53.6 | 60.3 | 57.6 |
| Health and social work | **61.3** | 50.1 | 44.8 | 47.5 | 62.9 | 52.7 | 58.5 | 60.6 | 61.8 | 49.5 | 57.8 | 62.9 | 59.1 | 55.4 | 45.9 | 67.9 | 60.4 | 50.9 | 35.3 | 60.0 | 63.7 | 50.7 | 59.5 | 55.9 | 49.6 | 46.8 | 55.9 | 61.6 |
| Other community, social, and personal services | **58.1** | 42.0 | 40.6 | 40.2 | 59.4 | 44.7 | 56.2 | 52.5 | 54.5 | 40.0 | 41.6 | 56.2 | 70.2 | 67.4 | 47.5 | 57.1 | 62.0 | 68.0 | 34.7 | 60.6 | 20.8 | 32.8 | 54.4 | 50.2 | 53.2 | 41.4 | 45.8 | 42.8 |

The highlighted results are of significant value.

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
