# Peer review of "The GHG Emissions Generating Capacity by Productive Sectors in the EU: A SAM Analysis"

_sustainability, doi:10.3390/su13042363_

Round 1

Reviewer 1 Report

  1. The reviewed manuscript is comprehensive information and describes a study reporting the GHG emissions from production sectors in European Union countries.
  2. The title is not too long and reflects the content of the manuscript and conveys to the readers the goal of the research. 
  3. The abstract summarize the aim of the research concerning GHG emission in the EU-27, how it was conducted by Social Accounting Matrices, the major results and the final conclusion.

  1. The “GHG emissions” should be add to the keywords because is the main subject of the study.

  1. In the Methods section clearly describe the procedure to evaluate the generating capacity of GHG emissions by using emission multipliers and Social Accounting Matrices (SAMs).

  1. A large number of productive sectors (27) were considered in each country and data from the all tables are very interesting and easy to follow and understand.

  1. My suggestion is to comment in few words the reason why in some countries in Transport sector for example the capacity of GHG emissions generation is lowest or higher than in the other countries.
  2. The Conclusion section is well argued with discussion and recommendation and obviously as the study results show each EU-27 country must concentrate its own effort to decrease decarbonisation in all productive sectors.
  3. Authors (Barun and Pohit (2014), Barun et al. (2015); (Rose and Chen, 1991) are not found in the References List.
  4. Merge the number 3 with 4, eight (8) with 9 and 30 with 31 from the Reference list (doi is split from the paper title!)
  5. References no 11 and 12 must be write in both situations with lowercases or to capitalize the „de” word.
  6. The following Authors are not cited in the manuscript text:

  1. Pal, B.D, Ojha, V.P., Pohit S. and Roy,P. (2015). Impact of economic growth on Greenhouse Gas (GHG) emissions—Social Ac-43 counting Matrix (SAM) Multiplier Analysis. In book: GHG emissions and economic growth: A computable general equilib-44 rium model based analysis for India, Springer. doi.org/10.1007/978-81-322-1943-9_4 45
  2. Pal, B.D. and Pohit, S. (2014). Environmentally extended social accounting matrix for climate change policy analysis for India. 46 Journal of Regional Development and Planning 3(1). 61-75

Author Response

We are very grateful for your effort on revising the paper and your positive comments. We have modified accordingly the document and you can find below the concrete responses. In addition, the main changes in the paper are highlighted in yellow.

  1. The reviewed manuscript is comprehensive information and describes a study reporting the GHG emissions from production sectors in European Union countries.
  2. The title is not too long and reflects the content of the manuscript and conveys to the readers the goal of the research. 
  3. The abstract summarizes the aim of the research concerning GHG emission in the EU-27, how it was conducted by Social Accounting Matrices, the major results and the final conclusion.
  4. The “GHG emissions” should be add to the keywords because is the main subject of the study.
  5. In the Methods section clearly describe the procedure to evaluate the generating capacity of GHG emissions by using emission multipliers and Social Accounting Matrices (SAMs).
  6. A large number of productive sectors (27) were considered in each country and data from the all tables are very interesting and easy to follow and understand.

Thank you very much for your positive opinion of the paper.

  1. My suggestion is to comment in few words the reason why in some countries in Transport sector for example the capacity of GHG emissions generation is lowest or higher than in the other countries.

A brief comment regarding this issue has been included now in the text:

“In general, the differences in the emission generation capacity of the same sector in two countries are due to the confluence of several factors, such as different production functions, different mix of inputs and different productive processes. The countries mainly depend on their technological development, but also on their environmental legislation and implementation, among other issues.”

  1. The Conclusion section is well argued with discussion and recommendation and obviously as the study results show each EU-27 country must concentrate its own effort to decrease decarbonisation in all productive sectors.

Thanks again for your positive evaluation.

  1. Authors (Barun and Pohit (2014), Barun et al. (2015); (Rose and Chen, 1991) are not found in the References List.

This has been corrected in the new version of the paper. Actually, the two first references were already included, but in a wrong way.

  1. Merge the number 3 with 4, eight (8) with 9 and 30 with 31 from the Reference list (doi is split from the paper title!)

This has been corrected.

  1. References no 11 and 12 must be write in both situations with lowercases or to capitalize the „de” word.

Now, both situations are written with lowercases.

  1. The following Authors are not cited in the manuscript text:

Pal, B.D, Ojha, V.P., Pohit S. and Roy,P. (2015). Impact of economic growth on Greenhouse Gas (GHG) emissions—Social Accounting Matrix (SAM) Multiplier Analysis. In book: GHG emissions and economic growth: A computable general equilibrium model based analysis for India, Springer. doi.org/10.1007/978-81-322-1943-9_4 45.

Pal, B.D. and Pohit, S. (2014). Environmentally extended social accounting matrix for climate change policy analysis for India. Journal of Regional Development and Planning, 3(1), 61-75.

Both references were already cited in the previous version, but in a wrong way (actually, we had written Barun instead of Pal (Barun Pal) when citing in the text.

Reviewer 2 Report

In this paper, authors have tried to improve analyses on GHG emissions in Europe by using Social Accounting Matrices which is not commonly considered method when such analyses are conducted.

In my opinion, this idea is worth considering but I have some major issues regarding presentation of it in this paper.

I shall address those issues point by point.

  1. Title is somewhat misleading since it mentions EU-27. It is obvious already from the Abstract that authors included EU-27 countries as it was in 2010. However, EU also has 27 member states today but those are not the same as 27 referred in the paper (with Croatia as a new member state and United Kingdom as a former member state). In my opinion, it should be clarified in the title by either adding the year or in some other way.
  2. In the 6th line of introduction, temperature is marked incorrectly. It should be “°C”.
  3. In the section “2.2. The model”, it is said that “…the empirical application is performed with the SDA technique applied to sectoral production. It will allow us to disaggregate energy intesitite in the EU-27…” However, explanation how will this application allow that is missing here.
  4. I have noticed widespread use of footnotes in the text. I am not against footnotes if it is OK with the journal guidelines. However, I would prefer that the authors include in the text at least 5th footnote which contains mathematical expression.
  5. Results are well presented in the “Result” section but the section “Discussion” is completely missing which has a major negative impact on the quality of this paper. It should be either thoroughly included in the “Result” section (which in that case should be renamed to “Results and discussion”) or in the separate section named “Discussion”.
  6. In the “Conclusion and policy recommendations” section, policy recommendations are general. That is partly because the problems vary between member states but this is also connected with only basic interpretation and lack of discussion. With the improvements noticed in the 5th point, 6th point will also need to be rewritten.

As a recommendation to the editor, I would reconsider this paper for publication only if major and thorough revision be made in this manuscript.

Author Response

We are very grateful for your efforts in revising our paper and your accurate and useful comments. We have include them all in this new version of the paper. You can find below the replies to your comments. The main changes introduced in the manuscript are highlighted in blue.

  1. Title is somewhat misleading since it mentions EU-27. It is obvious already from the Abstract that authors included EU-27 countries as it was in 2010. However, EU also has 27 member states today but those are not the same as 27 referred in the paper (with Croatia as a new member state and United Kingdom as a former member state). In my opinion, it should be clarified in the title by either adding the year or in some other way.

Effectively, the use of EU27 can be confusing after Brexit. Therefore, we have removed the reference to the number of EU Member States in the title, and we have explained it in the abstract and the main text.

  1. In the 6thline of introduction, temperature is marked incorrectly. It should be “°C”.

It was a typo. Thanks for letting us know.

  1. In the section “2.2. The model”, it is said that “…the empirical application is performed with the SDA technique applied to sectoral production. It will allow us to disaggregate energy intensities in the EU-27…” However, explanation how will this application allow that is missing here.

A brief explanation has been added to the text:

“In this paper, the empirical application is performed with the SDA technique applied to sectoral production. The SDA allows a certain vector (or value) to be decomposed into a series of additive components. In the current case, we disentangle the vector of differences between emissions intensities generated by each MS and the EU-27 average into three main components, which aggregated resemble the total difference”.

  1. I have noticed widespread use of footnotes in the text. I am not against footnotes if it is OK with the journal guidelines. However, I would prefer that the authors include in the text at least 5thfootnote which contains mathematical expression.

As the editor has reminded us, the journal does not admit footnotes. We have introduce them all into the main text. They are highlighted in grey.

  1. Results are well presented in the “Result” section but the section “Discussion” is completely missing which has a major negative impact on the quality of this paper. It should be either thoroughly included in the “Result” section (which in that case should be renamed to “Results and discussion”) or in the separate section named “Discussion”.

The Results section has been rewritten and renamed as “Results and discussion”.

In the “Conclusion and policy recommendations” section, policy recommendations are general. That is partly because the problems vary between member states, but this is also connected with only basic interpretation and lack of discussion. With the improvements noticed in the 5th point, 6th point will also need to be rewritten.

Following the reviewer suggestion, “Conclusions” section has been also rewritten in order to be coherent with improved section “Results and discussion”.

Round 2

Reviewer 2 Report

I would like to thank the authors for replies and I believe that this article may be accepted for publication in present  form.